# Paw pressure and gait in middle-aged client-owned cats with and without naturally-occurring musculoskeletal disease

Nathalie Dowgray[1,2]*, Eithne Comerford[1,3,4], Alexander J. German[1,3], James Gardiner[1¤], Gina Pinchbeck[5], Karl T. Bates[1]

1 Institute of Life Course and Medical Sciences, University of Liverpool, Liverpool, United Kingdom, 2 International Cat Care, Tisbury, Wiltshire, United Kingdom, 3 School of Veterinary Science, University of Liverpool, Liverpool, United Kingdom, 4 The MRC-Versus Arthritis Centre for Integrated research into Musculoskeletal Ageing (CIMA), Liverpool, United Kingdom, 5 Institute of Infection, Veterinary and Ecological Sciences, University of Liverpool, Liverpool, United Kingdom

¤ Current address: Department of Sport and Exercise Sciences, Manchester Metropolitan University, Manchester, United Kingdom
* dowgray@liverpool.ac.uk

**Data Availability Statement:** All relevant data are within the manuscript and its Supporting Information files.

## Abstract

Musculoskeletal disease (MSD) is common in ageing cats, resulting in chronic pain and mobility impairment, but diagnosis can be challenging. We hypothesised that there would be differences between cats with and without MSD in paw pressure and spatiotemporal and kinetic gait metrics. A cohort of 53 cats, aged between 7 and 10 years from the North West of the United Kingdom, underwent an orthopaedic examination and walked on a pressure sensitive walkway. Thirty-one of the cats (58%) were determined to be apparently-healthy, based on a normal orthopaedic examination and having no history of MSD, whilst the remaining 22 cats (42%) had findings consistent with MSD; 13/22 cats (59%) had multiple limb involvement, 7/22 (32%) had forelimb involvement and 2/22 (9%) had hindlimb involvement. Bodyweight ($P = 0.048$) and body condition score (BCS; $P = 0.015$) were both greater in cats with MSD (mean bodyweight 5.4 ± 1.35 kg; median BCS 6, IQR 6–7.75) compared with apparently-healthy cats (mean bodyweight 4.7 ± 0.94 kg; median BCS 5, IQR 4.5–6.5). There was a relatively large intra-cat variation in spatiotemporal and kinetic gait variables (coefficient of variation >3.0%), whilst a linear mixed-effects model suggested no significant difference in spatiotemporal or kinetic gait variables between apparently-healthy cats and those with MSD. Palmar and plantar pressure asymmetry was assessed by pedobaro-graphic statistical parametric mapping (pSPM) within each individual cat, with no significant difference ($P = 0.353$) between the apparently heathy cats and those with MSD as to the presence or absence of asymmetry. Given the marked intra-cat variation and the 'multi-limb' nature of MSD in this cohort, it was not possible to differentiate healthy cats from those with MSD based on spatiotemporal and kinetic gait metrics or paw pressure asymmetry. Future work should examine gait in cats with defined musculoskeletal disorders (e.g. hip dysplasia) and also to track longitudinal changes within individual cats to better establish age-related trends.

**Funding:** The study was funded by a grant from Royal Canin. The sponsor collaborated on the design of the study and reviewed the manuscript for quality purposes prior to submission. However, they were not involved in data collection, clinical assessments, data analysis, interpretation, manuscript writing or the decision to submit the work for publication.

**Competing interests:** AJG is an employee of the University of Liverpool, but his post is financially supported by Royal Canin, a division of Mars Petcare. AJG has also received financial remuneration for providing educational material, speaking at conferences, and consultancy work from this company; all such remuneration has been for projects unrelated to the work reported in this manuscript. At the time the study was performed, ND was undertaking a post-graduate studentship funded by Royal Canin. Since October 2020, ND has been employed by International Cat Care, but also holds a part-time post-doctoral research position at the University of Liverpool, funded by Royal Canin. This does not alter our adherence to PLOS ONE policies on sharing data and materials.'

## Introduction

Musculoskeletal disease (MSD) is common in ageing cats, with degenerative joint disease (DJD), encompassing osteoarthritis (OA), being most prevalent [1], albeit depending on the methodology of the study and the population assessed [2]. Radiographic studies have estimated the prevalence of DJD for cats to be between 22% and 92% [3–7]; in contrast, a prevalence of only 2% was suggested in a study using non-radiographic data from primary care veterinary practices [8]. These differences highlight the difficulty in diagnosing feline DJD in clinical practice and suggest significant under-reporting.

Diagnosis of MSD in cats is commonly made using a combination of owner-observed changes, veterinary orthopaedic examination findings and radiography [9]. Although owner-observed changes in mobility, activity, grooming, temperament and response to analgesia can be useful in assessing chronic pain caused by DJD [7, 10–12], such observations are subjective. Feline DJD-specific clinical metrology instruments have been created to standardise these subjective measures [12–15], although limitations remain including the tendency for responses to be overestimated when they are used as the sole outcome measure [16]. Therefore, there is a need for more objective measures to improve the diagnosis of age-related MSD in cats such as gait analysis as used in other species [17–19].

Normal feline gait in healthy cats has been investigated with pressure-sensitive walkways, which measure paw pressure, vertical ground reaction forces and spatiotemporal gait kinematics [20–25]. Gait analysis has also been used to assess parameters in cats post-onychectomy; in cats with coxofemoral OA and stifle OA following cranial cruciate injury; in cats that have undergone femoral head and neck ostectomy; and also to detect hindlimb lameness [26–30]. Furthermore, gait metrics have been used to quantify post-operative lameness due to different surgical techniques [31], and also the effect of analgesia in post-surgical models [32] and naturally occurring OA [33]. A limitation of all these studies is that only cats with either a single condition or unilateral disease have been studied; since feline DJD is often bilateral, affecting multiple limbs and joints, the effects on gait, including any asymmetry are likely to be more complex [6, 7]. Given the limited information currently available, the aim of the current study was to use a pressure-sensitive walkway to measure gait metrics in senior pet cats with and without MSD. We hypothesised that cats with and without MSD would differ in paw pressure and spatiotemporal and kinetic gait metrics.

## Materials and methods

### Eligibility criteria

Client owned cats that had been enrolled in the Cat Prospective Ageing and Welfare Study (CatPAWS), between 2017 and 2020 and meeting all recruitment and eligibility criteria [34], were eligible for participation. Ethical approval for CatPAWS was granted by the University of Liverpool Veterinary Research Ethics committee (VREC491abcd). Briefly, the cats needed to be between 7 and 10 years of age and owners had to consent to visit the Feline Healthy Ageing Clinic, University of Liverpool, UK (the data collection clinic for CatPAWS) every six months. Additional eligibility criteria for the current study were that the enrolled cats had to have had an orthopaedic examination performed, and have completed two or more successful crossings of a pressure-sensitive walkway (see below) at either their enrolment or first annual appointment. A successful crossing was defined as the cat passing the whole length of the walkway in a straight line, at a continuous walk (<1m/s) with the head facing forward during the entire crossing [23].

## Examination for MSD

As part of their involvement in the CatPAWS study, all cats received regular clinical assessments with a qualified veterinarian and registered veterinary nurse, including bodyweight measurement, body condition scoring (BCS) and an orthopaedic examination performed by the veterinarian and based on previously-published criteria [35]. The orthopaedic examination included a visual assessment on cats' willingness to walk around the consultation room [9, 36] and an assessment of coat condition and grooming activity [7, 10]. Each cat was also assessed for muscular asymmetry, the vertebral column was palpated for evidence of pain, and range of movement (ROM) of the neck and tail was also checked. On a limb-by-limb basis, the digits were examined for thickening of the claws and evidence of pain on manipulation; then, working up the limb, the appendicular joints (carpus, elbow, shoulder, tarsus, stifle and coxofemoral) were palpated for thickening, ROM and pain. Pain assessment was based on previously-published systems and classified as follows: 'no resentment', 'tries to escape/prevent manipulation', 'bite/hiss', 'marked guarding of area' [35]. Thickening of joints was recorded as 'normal', 'mild-to-moderate' or 'severe', whilst ROM was recorded as 'normal', 'reduced' or 'cannot manipulate' [35]. The cats were held with gentle restraint throughout the examination and a temperament assessment was recorded at the end [35].

Cats were defined as 'apparently-healthy middle-aged cats' if they had no history of MSD and had no discernible issues on their orthopaedic examination. Cats were classified as 'having MSD' if they had a history of a disease involving the orthopaedic system, such as DJD, a femoral head and neck excision or a previous fracture, or findings consistent with MSD on their orthopaedic examination, including: pain upon joint manipulation, joint thickening, joint effusion and a reduction in the ROM of the joint [35, 37]. For analysis of paw pressure and spatiotemporal and kinetic gait metrics, cats in the MSD group were assigned to sub-groups based on the limb(s) affected. Additionally, all cats' limbs were labelled as healthy or not healthy for assessment (e.g., all apparently healthy cats recorded four healthy limbs, whilst the MSD group could have one or more non healthy limbs.).

## Gait analysis

A pressure-sensitive walkway, measuring 0.45m by 1.76m, was used for gait analysis (HRV3 High resolution walkway system, Tekscan, USA). Spatial resolution for this walkway was 3.9 sensel per $cm^2$ and the frame rate was set at 100 frames per second [38]. A modified, clear poly-tunnel was placed over the walkway to encourage cats to move in a straight line (Fig 1). Cats were encouraged to walk at their own pace across the walkway, either using positive reinforcement (food treats, verbal praise, stroking or grooming) or by providing access to a hiding place at the end of the tunnel (basket or cat hide). Each crossing was filmed using a digital camera (Panasonic Lumix DMC-TZ6) positioned perpendicular to the pressure mat.

At the start of each weekly session, the pressure-sensitive walkway was calibrated, involving the creation of a calibration file, which was then used for all sessions over the following 2–3 days [38]. Each of the 3 walkway sections was calibrated with an object of known weight (usually a bag of canine dry food or a plastic container containing cat litter, typical weight 10–16 kg), as per manufacturer recommendations (Tekscan). This calibration method was used for data collected between February 2017 and May 2019. However, interim data analysis revealed that paw pressure readings were greater than expected in some cats; after discussion with the manufacturer, a three-legged stool loaded with 10kg of weight plates (10.9 kg in total) was instead used for calibration, enabling pressure to be focused over three small areas, thereby mimicking the pressure generated by a feline paw [22, 39]. To enable analysis of all data, files collected between February 2017 and May 2019 were retrospectively calibrated using a stool calibration file created in June 2019. Analysis of retrospectively-calibrated longitudinal data

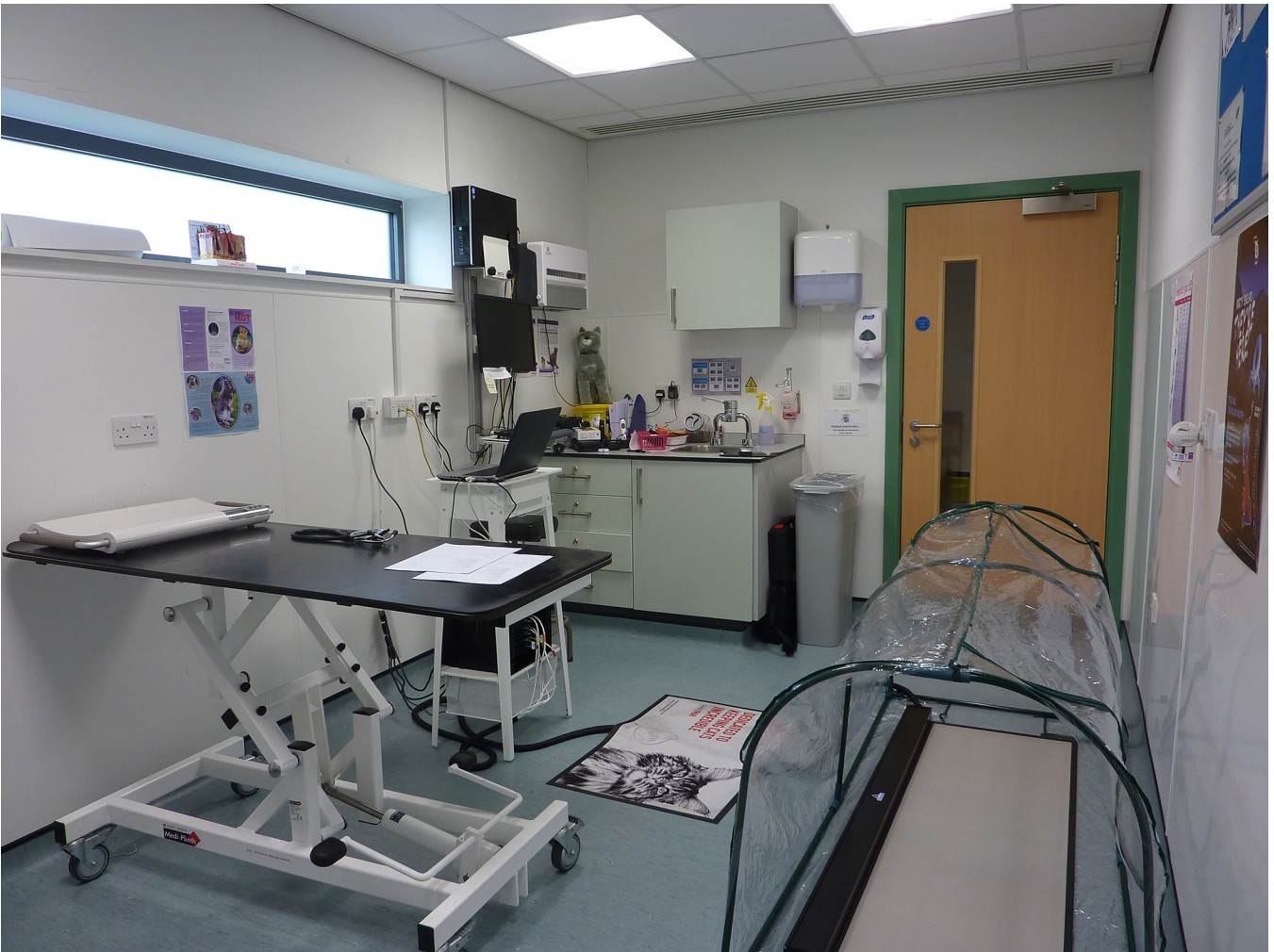

**Fig 1. Layout of the consultation room showing the pressure-sensitive walkway with the polytunnel.** The digital camera (Panasonic Lumix DMC-TZ6) used for filming was positioned on the middle shelf of the laptop stand (obscured from view in the photograph by the consulting room examination table) perpendicular to the direction of travel, providing a lateral view of the cat's gait.

showed that, over a two-year period there was also a decline in the sensitivity of the pressure-sensitive walkway, which could have been adjusted for using contemporaneous calibration but not by using retrospective calibration. Since the cycle time, impulse and bodyweight of the cat (mass) were all known, a calibration factor was created and applied to the ground reaction forces for analysis, as follows:

*Calibration factor = (Total impulse (Ns)/Cycle time (sec))/Cat's mass (N)* [38]

Gait parameters were extracted for each gait cycle using the software ('Walkway Research', version 7.66–03, Tekscan, USA) and those assessed for each limb are summarised in Table 1. Duty factor (stance time/stride time), normalised peak vertical force (PVF) and normalised vertical impulse (VI) were calculated. Normalised ground reaction forces were calculated by dividing the peak forces by the cat's bodyweight in Newtons. Fore- and hindlimb symmetry indices were calculated for duty factor (DF), PVF and VI as previously described [24, 25, 29, 30], where 0% represents perfect symmetry.

Pressure data for each crossing were exported as a comma-separated value files (.csv) and analysed using tools from the suite of pedobarographic statistical parametric mapping (pSPM)

**Table 1. Spatiotemporal and kinetic parameters extracted for each gait cycle and assessed for each limb.**

| Spatiotemporal parameters | | |
|---|---|---|
| *Variable* | *Description* | *Unit* |
| Stance time | Time from first contact to last contact of each paw | seconds |
| Swing time | Time between last contact of proceeding and next contact of two consecutive paw falls | seconds |
| Stride length | Distance between two posterior points of consecutive paw falls | metres |
| Stride velocity | Stride length divided by the stride time for each limb | metres/second |
| Duty factor (DF) | Percentage of time per cycle the paw spends on the ground in each limb cycle, calculated by dividing the stance time by the stride time | % |
| Velocity | Gait distance divided by the gait time | metres/second |
| Gait cycle time | Time to complete one gait cycle (second) | seconds |
| **Kinetic parameters** | | |
| *Variable* | *Description* | |
| Maximum PVF | Maximum force during the stance of the foot | Newton |
| Normalised PVF | PVF / Bodyweight in Newtons | percentage |
| Maximum VI | Area under the force time curve for the duration of the stance | Newton seconds |
| Normalised VI | VI / Bodyweight in Newtons | percentage |

Data collected using the software 'Walkway Research', version 7.66–03, Tekscan, USA. DF = Duty Factor, PVF = Peak Vertical force, VI = Vertical impulse

functions within MATLAB (Version 9.6, MathWorks ®, Natick, Massachusetts, USA) [40, 41]. Peak pressure data were extracted, and peak pressure images (p-images) created and labelled (left forelimb [LF]; right forelimb [RF]); left hindlimb [LH]; right hindlimb [RH]) for each crossing, resulting in the production of 2–3 p-images per limb. Left peak pressure images were flipped on a vertical axis, to enable direct comparisons to be made with right p-images. All p-images, for each cat and paw, were registered to each other in a vertical stack using a two-stage, rigid-body transformation via an algorithm that aimed to minimise the between-image mean square error and ensuring optimal overlap of homologous structures [40–42]. In this respect, the first print for each paw was initially used as a template, with all other prints aligned to this. Following alignment, a mean pressure record was created and used as the basis for a second registration [40–42]. In contrast to human foot pressure records, on which this process has been extensively tested [40–48], cat prints were round; as a result, all final images were visually assessed to ensure that cranial-caudal alignment on the vertical axis was maintained in the final mean image. To achieve this, the mean image was aligned against a grid of equivalent size to the pixels of the pressure-sensitive walkways, enabling fine manual adjustments to be made if required. Any such adjustments were applied to the p-image stack resulting in the creation of a new aligned mean. Once completed, a mean peak pressure p-image was created for each limb of each cat using their individual peak pressure records, the total number of p-images per cat limb is the p-image stack denominator.

## Statistical analysis

Statistical analysis was performed using an online statistical environment and language (R, version 4.0.0 [49]) with several additional packages as explained below. Gait parameters were first assessed for normality using histograms, Q-Q plots and the Shapiro-Wilk test. Frequencies

were reported as numbers and percentages, normally-distributed data were described using mean, standard deviation (SD), minimum (min) and maximum (max); data that were not normally-distributed were instead described using median and inter-quartile range (IQR). Parametric tests were used for normally-distributed data and non-parametric tests used where data were not normally distributed. The level of assumed statistical significance was $P < 0.05$.

**Individual cat analysis.** The following additional R packages were used in analyses using data from individual cats:"ggplot version 2 3.3.6" [50], "ggpubr version 0.4.0" [51] and "reshape version 0.8.9" [52]. Descriptive statistics, as described above, were created for all successful crossings from each cat. The PVF and VI distribution within each cat across all limbs were assessed by analysis of variance (ANOVA). These data were then compared with the clinical data (MSD status and orthopaedic examination findings) to determine if any relationship was present, for example asymmetry. As this exercise was for data visualisation only, *P*-values were not corrected for multiple comparisons. Between-group distribution was assessed using Fisher's exact test.

Asymmetry in peak pressure distribution was assessed by comparing p-images between LF and RF, and also between LH and RH, using topological *t*-tests from the pSPM suite in MATLAB [40–42, 46]. In traditional analytical approaches, statistical tests are typically conducted on single pressure values extracted from either discrete anatomical regions (e.g. mean or maximum 'heel' pressure) or the entire pressure record (e.g. mean or maximum pressure). However, in pSPM, data from the paw pressure records are smooth and continuous, with neighbouring pixels in pressure images being were neither biologically nor statistically independent [40–42, 46]. To address this, pSPM generates a continuous statistical map across the entire pressure record, with random field theory then used to conduct inference topologically, based on the height and size of connected clusters of pixels in the image that remained following suitably high SPM thresholding [53]. The final output, therefore, was the identification of individual or clusters of pixels that significantly differed in two samples of pressure images, and accounting for their two-dimensional topological characteristics. The two-sample students *t*-test was performed with the aligned image stacks to examine differences in pressure between a pixel and the surrounding pixels [41]. The presence or absence of left-right mean peak pressure asymmetry was determined for each individual cat, and this was then assessed between groups using Fisher's exact test, whilst Wilcoxon's signed ranks test was used to assess the effect of p-image stack denominator on determination of asymmetry.

The coefficient of variation (CV) was used to examine individual cat variation [24]; within-cat CV was calculated for all the gait parameters for each session, with a CV $\leq$ 3% considered to be 'low variability' [42, 54, 55].

**Comparing gait parameters in apparently-healthy middle-aged cats and those with MSD.** The following additional R packages were used to compare gaits between groups of cats and limbs with and without MSD.; "sp version 1.4.7" [56], "Rcpp version 1.0.8.3" [57], "raster version 3.5.15" [58], tidyverse version 1.3.2" [59], "rstatix version 0.7.0" [60] and "epitools version 0.5.10.1" [61]. Between-group comparisons of age, bodyweight, BCS, velocity and symmetry indices were made with Student's *t*-test or Wilcoxon's signed ranks test. The effect of sex on the presence of MSD was also assessed by calculating the odds ratio and expressed with associated 95% confidence intervals (95%-CI).

Spatiotemporal gait and ground reaction force data were analysed with linear mixed-effects models, using R packages 'lme4 version 1.1.29' [62] and 'lmerTest version 3.1.3' [63]. Velocity, sex, BCS, weight and musculoskeletal health category were included as fixed effects, individual cat was included as a random effect and assessed using the population variance ($o^2$). Given the many potential fixed effects, bidirectional elimination was performed, with forwards elimination (using a significance of $P < 0.1$) being used to create an initial model for each gait

parameter; subsequently, sequential backwards elimination was undertaken where the least significant variable (using a significance of P<0.05) was removed over repeated refinements until the best fit model was found (based on Bayesian information criterion [BIC], where the model with the smallest value BIC was assumed to have the best fit). Residual plots were then assessed to ensure that models model met the assumptions of linearity and homoscedasticity. Results are reported as estimates of the regression coefficients (β) with associated 95%-CI.

## Results

### Study cohort

Between 2017 and 2020, 211 cats were enrolled in CatPAWS, 53 of which (23 female, 30 male; all neutered) were eligible for inclusion in the current study. The remaining cats did not meet the inclusion criteria of ≥2 successful crossings of the pressure mat or a complete orthopaedic analysis. Descriptive statistics for individual apparently healthy and MSD cats are summarised in Table 2. Raw spatiotemporal and kinetic parameters from all 53 cats are available in S1 Data set.

Median age of the cats was 8 years (IQR 7–9 years), and mean weight was 4.9 kg (SD 1.17 kg). Thirty-one of the 53 (58%) cats were apparently healthy, whilst orthopaedic examination consistent with MSD were present in the remaining 22 cats (42%) cats. Thirteen (59%) of the 22 cats with MSD had multiple limb involvement (both fore and hindlimb involvement), 7 (32%) had forelimb involvement (2 cats unilateral RF, 2 cats bilateral and 3 cats unilateral LF) and 2 (9%) had bilateral hindlimb involvement. At the level of the individual limb per gait cycle, there were 706 'healthy' forelimbs and 254 with MSD available for assessment; there were also 686 'healthy' hindlimbs and 274 with MSD for assessment. In the apparently-healthy group, 15 (48%) and 16 (52%) and of the cats were male and female, respectively; in the MSD group, 15 (68%) and 7 (32%) of the cats were male and female, respectively. The odds of a male cat having MSD was 1.5 (95% CI 0.7–3.1; P = 0.360).

There were no significant differences in age at time of gait analysis (P = 0.940), velocity (P = 0.992), cycle time (P = 0.754) or correction factor (P = 0.486) between apparently healthy and MSD groups. However, bodyweight (P = 0.048) and BCS (P = 0.015) were greater in cats with MSD (5.4kg [1.35]; BCS 6 [IQR 6–7.75]) compared with apparently-healthy cats (4.7 kg [0.94]; BCS 5 [IQR 4.5–6.5]). Male cats (5.5kg [1.13]) were heavier than female cats (4.4kg [0.91]; P<0.001) but there was no between-sex difference in BCS (male cats 6 [IQR 5–7]; female cats 6 [IQR 5–7], P = 0.521), or velocity (male 0.71m/s [0.162]; female 0.66m/s [0.117]; P = 0.220). Leg length measurements were available from 45/53 (85%) cats. There was no significant difference in leg lengths between the groups (P = 0.125).

### PVF and VI distribution

Across the cohort, the expected pattern of greater forelimb PVF (cf., hindlimb) was seen in 31/53 cats (58%), with no left-right asymmetry, whilst a similar pattern was seen in VI measures in 34/53 cats (64%) [23]. All individual cat results for PVF and VI distribution are included in the supporting information (S1 File). There was no significant between-group difference in the distribution of PVF (P = 0.096) or VI (P = 1.000). Fig 2 illustrates the typical ground reaction force pattern in one apparently healthy cat with greater PVF and VI in the forelimbs compared with hindlimbs, but no left-right asymmetry.

### Pressure data

The number of p-images available varied amongst cats, with a median per limb of 8 (IQR 6–11); however, there were two apparently-healthy cats outliers, with 28 and 29 prints per

**Table 2. Summary of characteristics and gait parameters from 53 mature cats, expressed as mean (standard deviation) or median and interquartile range.**

| | Apparently Healthy (n = 31) | | Musculoskeletal disease (n = 22) | |
|---|---|---|---|---|
| *Bodyweight (kg)* | 4.74 | (0.942) | 5.41 | (1.352) |
| *Age (years)* | 8 | 7–9 | 8.1 | (1.06) |
| *BCS* | 5 | 4.5–6.5 | 6 | 6–7.75 |
| *Leg length (m)* | 0.32 | 0.32–0.34 | 0.32 | 0.29–0.33 |
| *Velocity (m/sec)* | 0.69 | (0.135) | 0.69 | (0.161) |
| *Gait cycle time (sec)* | 0.66 | 0.62–0.82 | 0.73 | (0.127) |
| *Correction factor* | 1.02 | (0.097) | 1.00 | (0.100) |
| ***Left Forelimb*** | | | | |
| *Stance Time (sec)* | 0.43 | 0.39–0.52 | 0.48 | 0.39–0.52 |
| *Swing Time (sec)* | 0.25 | 0.23–0.29 | 0.26 | 0.23–0.28 |
| *Duty factor (%)* | 0.64 | (0.035) | 0.64 | (0.023) |
| *Stride Length (m)* | 0.48 | (0.041) | 0.48 | (0.042) |
| *Stride Velocity (m/sec)* | 0.70 | (0.146) | 0.69 | (0.162) |
| *Peak Vertical Force (N)* | 30.14 | (8.597) | 32.52 | (10.257) |
| *Normalised PVF* | 0.63 | (0.072) | 0.61 | (0.058) |
| *Vertical Impulse (Ns)* | 10.31 | (3.666) | 11.05 | (4.323) |
| *Normalised VI* | 0.21 | (0.050) | 0.21 | (0.038) |
| ***Right Forelimb*** | | | | |
| *Stance Time (sec)* | 0.43 | 0.37–0.52 | 0.47 | 0.40–0.54 |
| *Swing Time (sec)* | 0.25 | 0.23–0.29 | 0.25 | 0.23–0.28 |
| *Duty factor (%)* | 0.64 | (0.035) | 0.65 | (0.037) |
| *Stride Length (m)* | 0.48 | (0.044) | 0.48 | (0.040) |
| *Stride Velocity (m/sec)* | 0.71 | (0.147) | 0.70 | (0.172) |
| *Peak Vertical Force (N)* | 29.63 | (8.208) | 33.01 | (11.278) |
| *Normalised PVF* | 0.62 | (0.064) | 0.61 | (0.075) |
| *Vertical Impulse (Ns)* | 9.97 | (3.537) | 11.21 | (4.574) |
| *Normalised VI* | 0.21 | (0.043) | 0.21 | (0.041) |
| *SI Forelimbs Duty Factor (%)* | 0.35 | (1.781) | 0.54 | (1.462) |
| *SI Forelimbs PVF (%)* | 0.72 | (3.120) | 0.43 | (2.807) |
| *SI Forelimbs VI (%)* | 1.55 | (3.324) | 0.39 | (3.386) |
| ***Left Hindlimb*** | | | | |
| *Stance Time (sec)* | 0.44 | 0.37–0.50 | 0.46 | 0.38–0.51 |
| *Swing Time (sec)* | 0.30 | 0.28–0.34 | 0.31 | 0.28–0.32 |
| *Duty factor (%)* | 0.60 | (0.024) | 0.61 | (0.034) |
| *Stride Length (m)* | 0.50 | (0.056) | 0.49 | (0.047) |
| *Stride Velocity (m/sec)* | 0.70 | (0.158) | 0.69 | (0.158) |
| *Peak Vertical Force (N)* | 24.49 | (7.427) | 26.32 | (8.506) |
| *Normalised PVF (%)* | 0.51 | (0.075) | 0.49 | (0.052) |
| *Vertical Impulse (Ns)* | 7.80 | (3.066) | 8.54 | (3.686) |
| *Normalised VI (%)* | 0.16 | (0.034) | 0.16 | (0.031) |
| ***Right Hindlimb*** | | | | |
| *Stance Time (sec)* | 0.44 | 0.28–0.49 | 0.46 | 0.40–0.51 |
| *Swing Time (sec)* | 0.30 | 0.27–0.36 | 0.32 | 0.26–0.32 |
| *Duty factor (%)* | 0.60 | (0.026) | 0.61 | (0.038) |
| *Stride Length (m)* | 0.49 | (0.051) | 0.49 | (0.050) |
| *Stride Velocity (m/sec)* | 0.70 | (0.148) | 0.69 | (0.163) |

*(Continued)*

**Table 2.** (Continued)

| | Apparently Healthy (n = 31) | | Musculoskeletal disease (n = 22) | |
|---|---|---|---|---|
| *Peak Vertical Force (N)* | 24.59 | (7.843) | 26.85 | (8.889) |
| *Normalised PVF (%)* | 0.51 | (0.081) | 0.50 | (0.059) |
| *Vertical Impulse (Ns)* | 7.67 | (2.995) | 8.64 | (3.750) |
| *Normalised VI (%)* | 0.16 | (0.031) | 0.16 | (0.034) |
| *SI Hindlimbs Duty Factor (%)* | 0.38 | (2.119) | 0.09 | (2.626) |
| *SI Hindlimbs PVF (%)* | 0.04 | (3.110) | 0.89 | (2.597) |
| *SI Hindlimbs VI (%)* | 0.76 | (3.641) | 0.53 | (2.860) |

Leg length data is incomplete and was available for n = 27 apparently healthy cats and n = 18 cats with musculoskeletal disease. BCS = body condition score, PVF = peak vertical force, VI = Vertical impulse, sec = seconds, m = meters, N = Newton's, Ns = Newton seconds, SI = Symmetry index

limb, respectively. A representative example of an apparently-healthy cat forelimb and hindlimb pressure distribution is shown in Fig 3, whilst mean p-images for all cats are included in the supporting information (S1 File). Twelve of 31 cats (39%) and 5/22 cats (23%) in the apparently-healthy and MSD groups, respectively, showed asymmetry between the left and right on either their fore or hindlimbs, but no significant group differences in the presence of intra-cat pressure asymmetry were evident (*P* = 0.353). Asymmetry was mainly detected in at the peripheral edges of the paw in both groups (10/12 apparently healthy; 4/5 MSD) and restricted to small numbers of pixels (1–3 out of a potential 36); a representative example is shown in Fig 4. Fewer p-images were available (12 [8–12] vs. 19 [13.5–24.5], *P*<0.001) from cats where asymmetry was identified, compared with those without asymmetry. Individual cat topological student's *t*-test results are included in the supporting information (S1 File).

## Comparing gait parameters in apparently-healthy middle-aged cats and those with MSD

No significant group differences were evident in symmetry indices for either fore- or hindlimbs (Table 3). The MSD group was categorised in two ways for analysis. Firstly, by cat, multi-limb (n = 13), RF (n = 2), LF (n = 3), bilateral fore (n = 2) and bilateral hind (n = 2). As these groupings contained small numbers of cats, those with affected forelimbs were also grouped together (n = 7) and the analysis repeated. An additional analysis, examining individual limbs, was also performed. To recognise the compensatory nature on gait that disease in one limb may have on the remaining limbs, all 'healthy' limbs from MSD classified cats were removed from the data set, leaving 580 healthy fore- and hindlimbs left in the comparison to 254 forelimbs and 274 hindlimbs with MSD.

Regardless of categorisation (individual limb or apparently healthy vs MSD), the presence of MSD had no significant effect on any gait parameters once body weight was included in the model and were not included as fixed effects in any of the final linear mixed-effects models following backwards elimination using BIC (Table 4). Further, there was no significant effect of BCS or sex on any of the gait parameters and so neither of these variables were included in the final models. However, as expected, increasing velocity was associated with swing and stance time and increased stride length across all limbs (*P*<0.001 for all).

Duty factor was significantly negatively associated with increasing velocity in the LF limb (β -0.053, 95%CI -0.086, -0.012; *P*<0.001) but not in the RF (β -0.040, 95%CI -0.081, -0.001; *P* = 0.059), LH (β -0.004, 95%CI -0.029, 0.030; *P* = 0.98) or RH (β -0.027, 95%CI -0.060, 0.006; *P* = 0.114). Inclusion of bodyweight significantly improved the fit of the final model for the

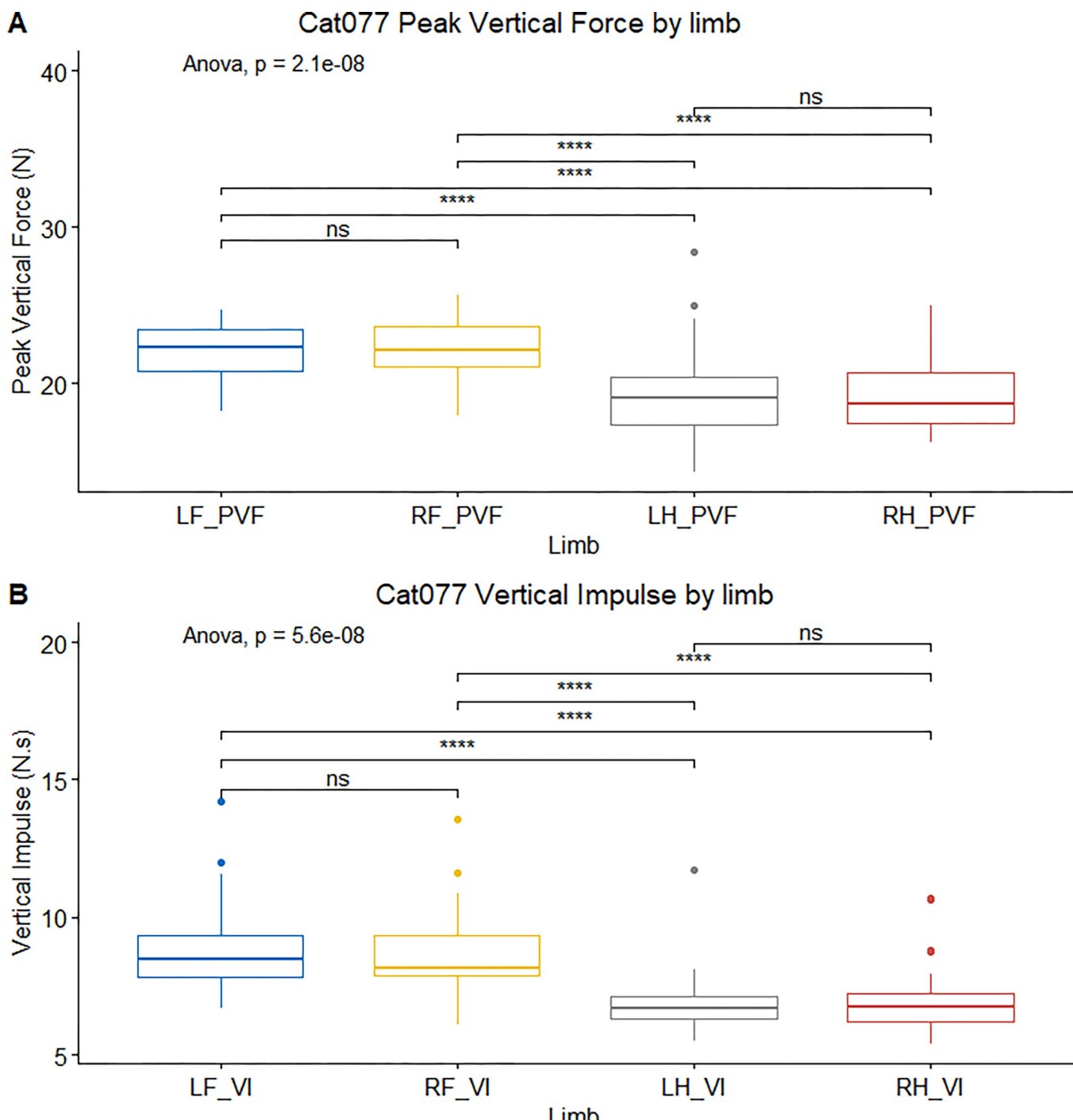

**Fig 2. Box-plot illustrating analysis of variance (ANOVA) of peak vertical force (PVF) and vertical impulse (VI) for individual Cat077. A)** Box-plot illustrating analysis of variance (ANOVA) of peak vertical force (PVF) by limb for individual Cat077 from 29 paw prints collected over 10 crossings. Forelimb PVF was greater than the -hindlimbs PVF, but there was no left-right asymmetry. **B)** Box-plot illustrating ANOVA of vertical impulse (VI) by limb for individual Cat077 from 29 prints collected over 10 crossings; again, forelimb VI was greater than hindlimb VI but there was no left-right asymmetry.

duty factor outcome variable in the left hindlimb ($\beta$ 0.012, 95%CI 0.007, 0.018; *P*<0.001). However, there was no significant effect of bodyweight on duty factor for any other limb and it was not included in any of these final models. The population variance ($o'^2$) for swing time,

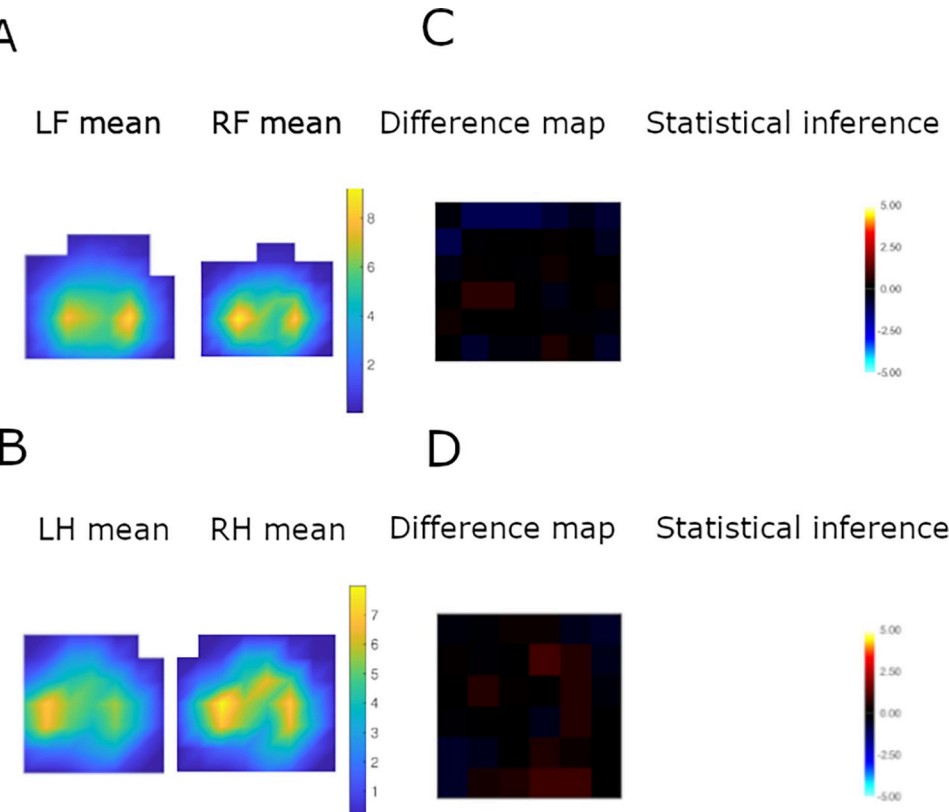

**Fig 3.** An illustrative example of mean palmar (A) and plantar (B) pressure distribution and of topological *t*-tests in palmar (C) and plantar (D) pressure distribution in the paws from Cat077. Mean images were created from 29 prints per limb and are orientated to the cranial aspect of the paw print to the right side, and the medial aspect to the top of the image. In the difference maps (C&D), red-yellow pixels indicate greater pressures in the left compared with right paws, whilst blue pixels indicate greater pressures in the right compared with left paws, black areas indicate no significant difference. Both palmar (A) and plantar (B) surfaces have two high-pressure areas in both the cranial and caudal sections of the paw surface. There were no statistically significant differences in paw pressure distribution between both the left and right forelimbs (C) and hindlimbs (D) as indicated by the predominately black colouring on the difference map.

stance time, stride length and duty factor were all small (<0.01) indicating little variation amongst the different measurements.

Peak vertical force (PVF) was associated with greater velocity and bodyweight (both *P*<0.001), the association to bodyweight was lost when PVF was normalised. Population variance ($o^{2}$) was large in all limbs (ranging from 6.63–7.55). Furthermore, VI was negatively associated with greater velocity (*P*<0.001) and positively associated with greater bodyweight (*P*<0.001), whilst $o^{2}$ was moderate (range 2.01–2.39 in all limbs). As with PVF, the significant association with bodyweight was lost when VI was normalised. Population variance ($o^{2}$) was low (<0.01) when both PVF and VI were normalised to bodyweight.

## Variation in gait parameters within apparently-healthy middle-aged cats and within those with MSD

The CV was assessed for all gait parameters and is presented in the supporting information (S1 and S2 Tables). For both groups (apparently healthy and MSD cats), there was large variation (>3.0%) in all parameters. The least variation was seen with stride length (CV range 3.7–4.9% across both groups) and force normalised to bodyweight (range across both groups 9.9–11.8%).

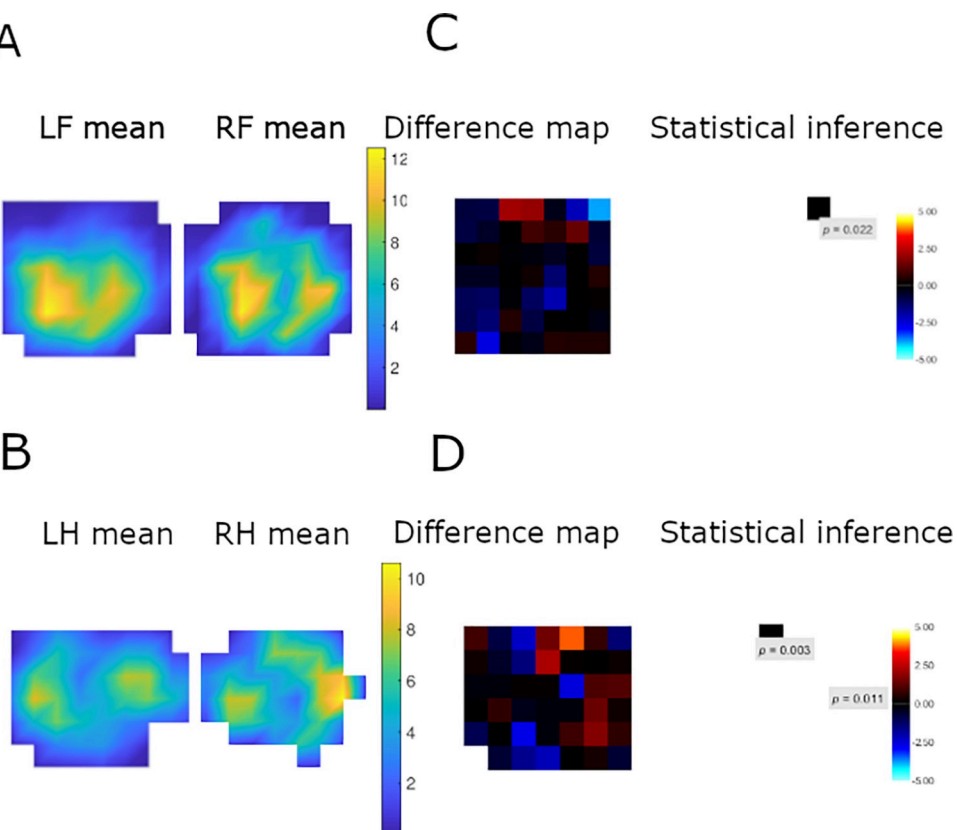

**Fig 4.** An illustrative example of mean palmar (A) and plantar (B) pressure distribution and of topological *t*-tests in palmar (C) and plantar (D) pressure distribution in the paw from Cat023. Images were created from 4 prints per limb and are orientated to the cranial aspect of the paw print to the right side, and the medial aspect to the top of the image. In the difference maps, red-yellow pixels indicate greater pressures in the left compared with right paws, whilst blue pixels indicate greater pressures in the right compared with left paws, black areas indicate no significant difference. Both palmar (A) and plantar (B) surfaces have two high-pressure areas in both the cranial and caudal sections of the paw surface. (C) In the palmar surface, there is a significant difference in the paw pressure distribution between the left and right forelimb. In the cranio-medial area the light blue pixel indicates the right forelimb has significantly higher pressure (*P* = 0.022) than the left forelimb in this area. D) In the plantar surface there is a significant difference in the paw pressure distribution between the left and right hindlimb. On the medial surface of the paw the left hindlimb had significantly higher pressure then the right hindlimb as indicated by the yellow pixel (*P* = 0.003) on the difference map and in the more central part of the plantar surface the right hindlimb has significantly higher pressure (*P* = 0.011) than the left hindlimb in this area as indicated by the lighter blue pixel.

**Table 3. Comparison of gait symmetry indices between 31 apparently-healthy cats and 22 cats with musculoskeletal disease.**

| | Forelimbs | | | Hindlimbs | | |
|---|---|---|---|---|---|---|
| | Apparently healthy | MSD | P value | Apparently healthy | MSD | P value |
| Duty factor | 0.35 (1.781) | -0.54 (1.462) | 0.961 | 0.38 (2.119) | 0.09 (2.626) | 0.167 |
| Peak vertical force | 0.72 (3.120) | -0.43 (2.807) | 0.590 | -0.04 (3.110) | -0.89 (2.570) | 0.475 |
| Peak Impulse | 1.55 (3.324) | -0.39 (3.386) | 0.910 | 0.76 (3.641) | -0.53 (2.860) | 0.424 |

Symmetry indices are expressed in mean and standard deviation, a value of 0.00 indicates perfect symmetry. For the statistical analysis, symmetry indices were all converted to positive numbers. MSD = Musculoskeletal disease.

**Table 4. Results from final linear mixed-effects models on spatiotemporal and kinetic gait parameters from 53 cats.**

| | Left Forelimb | | | Right Forelimb | | | Left Hindlimb | | | Right Hindlimb | | |
|---|---|---|---|---|---|---|---|---|---|---|---|---|
| | β | CI (95%) | P | β | CI | P | β | CI | P | β | CI | P |
| ***Swing time*** | | | | | | | | | | | | |
| (Intercept) | 0.367 | (0.348,0.386) | **<0.001** | 0.363 | (0.343,0.383) | **<0.001** | 0.400 | (0.371,0.430) | **<0.001** | 0.399 | (0.369,0.430) | **<0.001** |
| Velocity | -0.150 | (-0.180,-0.124) | **<0.001** | -0.144 | (-0.171,-0.118) | **<0.001** | -0.130 | (-0.170,-0.092) | **<0.001** | -0.124 | (-0.164,-0.085) | **<0.001** |
| **Random** | $o'^2$ | SD | | $o'^2$ | SD | | $o'^2$ | SD | | $o'^2$ | SD | |
| ID (intercept) | 0.000 | 0.019 | | 0.001 | 0.027 | | 0.001 | 0.039 | | 0.001 | 0.044 | |
| Residual | 0.000 | 0.025 | | 0.001 | 0.024 | | 0.001 | 0.036 | | 0.001 | 0.036 | |
| Observations | 198 | | | 198 | | | 198 | | | 198 | | |
| ***Stance time*** | | | | | | | | | | | | |
| (Intercept) | 0.791 | 0.750–0.834 | **<0.001** | 0.786 | 0.749–0.822 | **<0.001** | 0.685 | 0.635–0.736 | **<0.001** | 0.685 | 0.638–0.733 | **<0.001** |
| Velocity | -0.466 | (-0.522, -0.410) | **<0.001** | -0.464 | (-0.513,-0.416) | **<0.001** | -0.330 | (-0.399,-0.263) | **<0.001** | -0.334 | (-0.393,-0.270) | **<0.001** |
| **Random** | $o'^2$ | SD | | $o'^2$ | SD | | $o'^2$ | SD | | $o'^2$ | SD | |
| ID (intercept) | 0.002 | 0.048 | | 0.002 | 0.046 | | 0.003 | 0.054 | | 0.002 | 0.049 | |
| Residual | 0.003 | 0.054 | | 0.002 | 0.046 | | 0.004 | 0.067 | | 0.004 | 0.064 | |
| Observations | 198 | | | 198 | | | 198 | | | 198 | | |
| ***Stride Length*** | | | | | | | | | | | | |
| (Intercept) | 0.392 | (0.373,0.412) | **<0.001** | 0.396 | (0.377,0.416) | **<0.001** | 0.419 | (0.397,0.440) | **<0.001** | 0.414 | (0.393,0.435) | **<0.001** |
| Velocity | 0.124 | (0.099,0.145) | **<0.001** | 0.118 | (0.094,0.142) | **<0.001** | 0.106 | (0.081,0.132) | **<0.001** | 0.112 | (0.088,0.137) | **<0.001** |
| **Random** | $o'^2$ | SD | | $o'^2$ | SD | | $o'^2$ | SD | | $o'^2$ | SD | |
| ID (intercept) | 0.001 | 0.033 | | 0.001 | 0.035 | | 0.002 | 0.045 | | 0.002 | 0.042 | |
| Residual | 0.001 | 0.023 | | 0.000 | 0.022 | | 0.000 | 0.022 | | 0.000 | 0.021 | |
| Observations | 198 | | | 198 | | | 198 | | | 198 | | |
| ***Duty factor*** | | | | | | | | | | | | |
| (Intercept) | 0.677 | (0.653,0.701) | **<0.001** | 0.668 | (0.638,0.699) | **<0.001** | 0.545 | (0.511,0.580) | **<0.001** | 0.623 | (0.598,0.647) | **<0.001** |
| Velocity | -0.053 | (-0.086,-0.021) | **<0.001** | -0.040 | (-0.081,-0.001) | 0.059 | -0.004 | (-0.029,0.030) | 0.981 | -0.027 | (-0.060,0.006) | 0.114 |
| Weight | | | | | | | 0.012 | 0.007–0.018 | **<0.001** | | | |
| **Random** | $o'^2$ | SD | | $o'^2$ | SD | | $o'^2$ | SD | | $o'^2$ | SD | |
| ID (intercept) | 0.000 | 0.020 | | 0.001 | 0.026 | | 0.000 | 0.013 | | 0.000 | 0.022 | |
| Residual | 0.001 | 0.036 | | 0.002 | 0.044 | | 0.001 | 0.035 | | 0.001 | 0.036 | |
| Observations | 198 | | | 198 | | | 198 | | | 198 | | |
| ***PVF*** | | | | | | | | | | | | |
| (Intercept) | -4.397 | (-8.028,0.762) | **0.021** | -6.123 | (-9.151,-2.485) | **0.028** | -6.227 | (-9.721,-2.729) | **0.001** | -7.162 | (-10.816,-3.50) | **<0.000** |
| Velocity | 9.095 | (7.121,11.072) | **<0.001** | 11.460 | (9.623,13.323) | **<0.001** | 10.268 | (8.423,12.121) | **<0.001** | 11.381 | (9.591,13.194) | **<0.001** |
| Weight | 5.670 | (5.010,6.330) | **<0.001** | 5.673 | (5.008,6.340) | **<0.001** | 4.762 | (4.125,5.399) | **<0.001** | 4.839 | (4.166,5.512) | **<0.001** |
| **Random** | $o'^2$ | SD | | $o'^2$ | SD | | $o'^2$ | SD | | $o'^2e$ | SD | |
| ID (intercept) | 7.031 | 2.652 | | 7.332 | 2.708 | | 6.630 | 2.575 | | 7.550 | 2.748 | |
| Residual | 7.872 | 2.806 | | 6.673 | 2.583 | | 6.800 | 2.608 | | 6.306 | 2.511 | |
| Observations | 480 | | | 480 | | | 480 | | | 480 | | |
| ***PVF.NM*** | | | | | | | | | | | | |
| (Intercept) | 0.481 | (0.448, 0.514) | **<0.001** | 0.448 | (0.418,0.478) | **<0.001** | 0.339 | (0.308,0.370) | **<0.001** | 0.338 | (0.307,0.369) | **<0.001** |
| Velocity | 0.198 | (0.153, 0.240) | **<0.001** | 0.243 | (0.204,0.282) | **<0.001** | 0.238 | (0.199,0.277) | **<0.001** | 0.245 | (0.207,0.284) | **<0.001** |
| **Random** | $o'^2$ | SD | | $o'^2$ | SD | | $o'^2$ | SD | | $o'^2$ | SD | |
| ID (intercept) | 0.003 | 0.056 | | 0.002 | 0.051 | | 0.003 | 0.054 | | 0.003 | 0.057 | |
| Residual | 0.004 | 0.059 | | 0.003 | 0.055 | | 0.003 | 0.055 | | 0.003 | 0.054 | |
| Observations | 480 | | | 480 | | | 480 | | | 480 | | |
| ***VI*** | Fixed effects; velocity and weight; random effects = cat | | | | | | | | | | | |
| (Intercept) | 5.570 | (3.685,7.450) | **<0.001** | 4.400 | (2.510,6.285) | **<0.001** | 2.113 | (0.570,3.648) | **0.009** | 1.845 | (0.297,3.383) | **0.023** |
| Velocity | -9.879 | (-10.925,-8.851) | **<0.001** | -8.954 | (-9.864,-8.054) | **<0.001** | -6.634 | (-7.628,-5.667) | **<0.001** | -6.023 | (-6.891,-5.181) | **<0.001** |

*(Continued)*

**Table 4.** (Continued)

| | Left Forelimb | | | Right Forelimb | | | Left Hindlimb | | | Right Hindlimb | | |
|---|---|---|---|---|---|---|---|---|---|---|---|---|
| | **β** | *CI (95%)* | *P* | **β** | *CI* | *P* | **β** | *CI* | *P* | **β** | *CI* | *P* |
| Weight | 2.307 | (1.965,2.648) | **<0.001** | 2.390 | (2.041,2.738) | **<0.001** | 2.058 | (1.785,2.330) | **<0.001** | 2.019 | (1.740,2.298) | **<0.001** |
| **Random** | o'² | SD | | o'² | SD | | o'² | SD | | o'² | SD | |
| ID (intercept) | 1.875 | 1.369 | | 2.037 | 1.427 | | 1.111 | 1.054 | | 1.258 | 1.122 | |
| Residual | 2.159 | 1.469 | | 1.580 | 1.257 | | 2.023 | 1.422 | | 1.439 | 1.199 | |
| Observations | 480 | | | 480 | | | 480 | | | 480 | | |
| *VI.NM* | | | | | | | | | | | | |
| (Intercept) | 0.348 | 0.332–0.363 | **<0.001** | 0.334 | 0.319–0.348 | **<0.001** | 0.247 | 0.233–0.261 | **<0.001** | 0.242 | 0.229–0.255 | **<0.001** |
| Velocity | -0.201 | (-0.220,0.182) | **<0.001** | -0.185 | (-0.202,0.168) | **<0.001** | -0.130 | (-0.148,0.112) | **<0.001** | -0.124 | (-0.140,0.107) | **<0.001** |
| **Random** | o'² | SD | | o'² | SD | | o'² | SD | | o'² | SD | |
| ID (intercept) | 0.001 | 0.030 | | 0.001 | 0.029 | | 0.000 | 0.022 | | 0.001 | 0.022 | |
| Residual | 0.001 | 0.026 | | 0.001 | 0.024 | | 0.001 | 0.026 | | 0.001 | 0.023 | |
| Observations | 480 | | | 480 | | | 480 | | | 480 | | |

Final results following backwards elimination using BIC to, individual cat included as a random effect. Observations refer to the total number of gait cycles (198) or ground reaction forces (480). PVF = Peak Vertical Force, PVF.NM = Peak Vertical Force normalised to bodyweight, VI = Vertical impulse, VI.NM = Vertical impulse normalised to bodyweight, β = regression coefficients, o'² = population variance, SD = standard deviation, ID = individual cat. Significant *P*-values are presented in bold.

## Discussion

The aim of the current study was to determine if cats with and without MSD would differ in paw pressure and spatiotemporal and kinetic gait metrics. This study has determined that these parameters could not differentiate cats suffering from MSD from cats that were apparently healthy. These findings are most likely to be the result of the large intra- and inter-cat variation seen (S1 and S2 Tables), and also variability in how MSD presented, not least the fact that multi-limb disease was common. Given the small number of cats assessed, especially those with unilateral disease, the lack of significant differences is not surprising. Where significant differences between healthy and MSD cats have previously been identified using gait analysis, the groups studied either a specific disease condition (e.g., coxofemoral joint arthritis [28]) or unilateral disease (e.g. stifle disease or post-surgical intervention [26, 27, 29, 30]). Intra-cat gait variability in this study, as measured by CV, was greater than previously reported [24], both in apparently-healthy and MSD groups. These findings suggest that this type of gait analysis may have limited utility in pet cats with naturally-occurring non-specific MSD. Longitudinal gait analysis, tracking changes in individuals, might prove to be a better approach for cats. Additionally, further work examining the range of motion of joints and limb placement would improve understanding of the impact of MSD in cats. In this regard, DJD is already known to be associated with a reduced range of motion as assessed by goniometry in cats [35]. Use of markers and video analysis of gait was considered during the conception of this study but was not included due to practicalities associated with cat compliance, the suitability of the laboratory space available for free roaming cats to be in and the wider data collection required for the CatPAWs conception [38].

Differences between the current study and previous research might also be the result of different methodology. In an earlier study [24], each cat undertook three trials using a longer walkway, enabling more gait cycles to be collected. Further, screening radiography was performed (including controls) as part of their orthopaedic examination [24]; the lack of screening radiography in the current study might have led to cats incorrectly being assigned to the apparently-healthy group.

This study confirms the need to control for the velocity of the cats whilst collecting gait parameters, indicating that pressure sensitive treadmills might be beneficial in circumstances where cross-sectional analysis across a population of cats is required. However, variation in velocity may be a useful parameter to assess with naturally occurring disease. Differences in speed between healthy and mobility-impaired dogs have been demonstrated [64], where dogs with mobility impairment had a naturally lower velocity, suggesting that velocity might be a useful variable for diagnosis in this species. Velocity did not differ significantly between groups in the current study, and it is difficult to explain why cats and dogs with limb mobility issues appear to be different in this respect. In previous research, dogs were led by their owners at a trot [64], whereas the cats in the current study could choose their own speed. It might be worthwhile assessing longitudinal changes in velocity in relation to the development of MSD, either via gait analysis or accelerometer data. (67) Ideally, speed (and other spatiotemporal parameters) should also be normalised to leg length [65]; however, this was not possible in the current study because this measurement was not taken in all cats. However, where this was recorded, no significant between-group difference was evident ($P$ = 0.125, Table 2), suggesting that differences in leg length were unlikely to have unduly influenced the speed and other spatiotemporal parameters.

Unlike dogs, cats use more three-dimensional space in their daily lives (e.g. through climbing) and changes in jumping behaviours have remained important within DJD clinical metrology instrument refinement [66, 67]. With this in mind, accelerometers are potentially a better tool to explore changes in activity with naturally occurring musculoskeletal disease in cats, however there is significant variation seen between cats [68] and to date their use has best been established with intervention type studies where the cat acts as their own control [33, 69]. Additionally, gait speed has been associated with ageing and cognitive performance in dogs [70], so exploring cognitive decline as a comorbidity with musculoskeletal disease in ageing cats should be considered when investigating changes in mobility and velocity in ageing cats.

In addition to the marked intra- and inter-cat variability in gait data, other study limitations include the small sample size, the resolution of the pressure mat used and limited number of gait cycles recorded for each cat. It has been suggested that a minimum of 400 steps is required to capture kinematic variability in humans walking on treadmills [71], although more recent work has suggested that there are only minor advantages to having >200 steps per subject in analyses of peak plantar pressure in humans [44]. Conversely, sample sizes of <20 might not accurately reflect the population mean and might be insufficient for assessing pixel-level variation using topological approaches such as pSPM [44]. The number of steps, per cat per limb, in this study varied from 4 to 29, which might have contributed to the degree of asymmetry detected using topological $t$-testing in both groups, as supported by the observed relationship between number of p-images and the presence of asymmetry. Furthermore, the greatest asymmetry was identified around the periphery of the paw, and this might simply reflect minor changes in overall contact area step-to-step, as well as minor inconsistencies in the p-image registration process.

Available sample size was also limited by the willingness of cats to walk across the pressure sensitive walkway, not least the cats with MSD. Furthermore, cats in the MSD group were significantly heavier than those in the apparently healthy group, which might partly be due to the greater proportion of male cats represented. Body condition score was also greater, suggesting a possible effect of increased adiposity, consistent obesity being a risk factor for MSD in cats [72]. As well as an increased mechanical load, the metabolic effects of obesity may create a pro-inflammatory state-and further exacerbate the clinical signs [73]. However, given that overweight adult cats are also less active [74], our results might have been influenced by selection bias, because orthopaedic examination was used as the only screening method for feline

MSD. Conversely, food was used as a reward, to encourage cats to cross the walkway, which might instead have selected for food motivated cats. Nonetheless, sex and BCS were not retained in any of the final models and bodyweight was not retained after normalisation. Therefore, increased bodyweight and BCS did not cause any detectable differences in the gait parameters assessed.

## Conclusions

In this study, cats with MSD had significantly greater bodyweight and BCS than apparently-healthy cats, but there were no significant differences in paw pressure and spatiotemporal or kinetic gait metrics. These results are likely to be the result of both high intra- and inter-cat variability, difficulties in collecting usable gait data from pet cats and the multi-limb nature of MSD in cats. Future work should examine gait in a larger population of cats with defined musculoskeletal disorders (e.g. hip dysplasia or osteoarthritis), and aim to capture more images per cat, either by increasing the number of pressure mat crossings or by using a longer pressure mat. Use of a walkway with greater resolution is also recommended for pSPM analysis. Finally, to determine age-related changes in gait, cats should be monitored longitudinally during their senior years.

## Supporting information

**S1 Table. Summary of co-efficient of variation in gait parameters from n = 31 apparently healthy middle-aged cats.** Min = minimum value, Max = maximum value, Q1 = 1st quartile, Q3 = 3rd quartile, SD = standard deviation. Values in bold indicate distribution.
(DOCX)

**S2 Table. Summary of co-efficient of variation in gait parameter from n = 22 middle-aged cats with musculoskeletal disease.** Min = minimum value, Max = maximum value, Q1 = 1st quartile, Q3 = 3rd quartile, SD = standard deviation. Values in bold indicate distribution.
(DOCX)

**S1 File. Analysis of variance (ANOVA) of peak vertical force (PVF) and vertical impulse (VI) by limb and mean palmar and plantar pressure distribution and topological t-tests for palmar and plantar pressure distribution in the paws, of 53 cats.**
(DOCX)

**S1 Data set. Raw spatiotemporal and kinetic parameters from 53 cats.** Group = Musculoskeletal disease (MSD) or Apparently Healthy (AH), Walk = gait cycle recorded, W(N) = cats weight in newtons at time of recording, LF, RF., LH, RH indicated limb recorded. MaximunForceBW = maximum force as a % of body weight, MaximunForceN = maximum force in newtons, FTIBW = maximum vertical impulse as a % of body weight, FTIM = maximum vertical impulse in newtons, MPP = maximum peak pressure in newtons per m$^2$, TTFI = maximum vertical impulse for all four limbs, other parameters are as outlined in Table 1 of the main manuscript.
(XLSX)

## Acknowledgments

Kelly Eyre RVN for her involvement in the investigation and assisting with gait data collection, Rebecca Fonseka for her assistance in reviewing and sorting the raw gait data, the staff at the University of Liverpool Veterinary Practice for allowing us to work in their premises and all the owners of the cats for attending the clinic.

## Author Contributions

**Conceptualization:** Nathalie Dowgray, Eithne Comerford, Alexander J. German, Karl T. Bates.

**Data curation:** Nathalie Dowgray, Karl T. Bates.

**Formal analysis:** Nathalie Dowgray, James Gardiner, Karl T. Bates.

**Funding acquisition:** Alexander J. German.

**Investigation:** Nathalie Dowgray.

**Methodology:** Nathalie Dowgray, Eithne Comerford, Karl T. Bates.

**Project administration:** Nathalie Dowgray, Alexander J. German.

**Resources:** Nathalie Dowgray, Eithne Comerford, Alexander J. German, Gina Pinchbeck, Karl T. Bates.

**Software:** Nathalie Dowgray, James Gardiner, Karl T. Bates.

**Supervision:** Eithne Comerford, Alexander J. German, Gina Pinchbeck, Karl T. Bates.

**Validation:** Karl T. Bates.

**Visualization:** Nathalie Dowgray.

**Writing – original draft:** Nathalie Dowgray.

**Writing – review & editing:** Eithne Comerford, Alexander J. German, James Gardiner, Gina Pinchbeck, Karl T. Bates.

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
