## [Decision Letter · Decision Letter 0]

21 Aug 2024

PONE-D-24-25503Paw pressure and gait in middle-aged client-owned cats with and without naturally-occurring musculoskeletal diseasePLOS ONE

Dear Dr. Dowgray,

Thank you for submitting your manuscript to PLOS ONE. After careful consideration, we feel that it has merit but does not fully meet PLOS ONE’s publication criteria as it currently stands. Therefore, we invite you to submit a revised version of the manuscript that addresses the points raised during the review process.

Please note that we have only been able to secure a single reviewer to assess your manuscript. We are issuing a decision on your manuscript at this point to prevent further delays in the evaluation of your manuscript. Please be aware that the editor who handles your revised manuscript might find it necessary to invite additional reviewers to assess this work once the revised manuscript is submitted. However, we will aim to proceed on the basis of this single review if possible. 

We look forward to receiving your revised manuscript.

Kind regards,

Vanessa Carels

Staff Editor

PLOS ONE

Journal Requirements:

2. Thank you for stating the following in the Competing Interests section: "AJG is an employee of the University of Liverpool, but his post is financially supported by Royal Canin, a division of Mars Petcare. AJG has also received financial remuneration for providing educational material, speaking at conferences, and consultancy work from this company; all such remuneration has been for projects unrelated to the work reported in this manuscript. At the time the study was performed, ND was undertaking a post-graduate studentship funded by Royal Canin. Since October 2020, ND has been employed by International Cat Care, but also holds a part-time post-doctoral research position at the University of Liverpool, funded by Royal Canin."

Reviewers' comments:

Reviewer's Responses to Questions

**Comments to the Author**

1. Is the manuscript technically sound, and do the data support the conclusions?

Reviewer #1: Yes

2. Has the statistical analysis been performed appropriately and rigorously? 

Reviewer #1: Yes

3. Have the authors made all data underlying the findings in their manuscript fully available?

Reviewer #1: Yes

4. Is the manuscript presented in an intelligible fashion and written in standard English?

Reviewer #1: Yes

5. Review Comments to the Author

Reviewer #1: This is an interesting study investigating if a pressure sensitive walkway can differentiate cats with MSD and without MSD. Methods are thoroughly written, and limitations are well discussed. I have only several minor comments on the manuscript, and they are below.

1. Table2. If the authors used the following formula to calculate the symmetry indices (from the cited papers), negative values should not be seen. SIXFz=abs((XFzFL-XFzFR)/(XFzFL+XFzFR))x100

2. The reviewer is not clear about the definition of “forelimb involvement”, “hindlimb involvement” and “multiple limb involvement”. Does forelimb involvement mean that only a single forelimb is affected? If bilateral forelimbs are affected, that cat falls into the “multiple limb involvement”?

6. PLOS authors have the option to publish the peer review history of their article (what does this mean?). If published, this will include your full peer review and any attached files.

Reviewer #1: No

---

## [Author Response · Author response to Decision Letter 0]

24 Aug 2024

Thank you for your helpful review, we hope are following amendments are satisfactory. 

Point 1: Apologies, I had forgotten to apply the absolute value to the formular, this has now been corrected. 

Point 2: Thank you for raising this point. I have clarified the groups a little more in the results (see below) and in doing so noticed a mistake in the main manuscript, one of the forelimb cases had been miss counted as being multilimb, this mistake has been corrected. This mistake did not affect the files being used for statistical analysis and will have no impact on the results. 

‘Thirteen (59%) of the 22 cats with MSD had multiple limb involvement (both fore and hindlimb involvement), 7 (32%) had forelimb involvement (2 cats unilateral RF, 2 cats bilateral and 3 cats unilateral LF) and 2 (9%) had bilateral hindlimb involvement.’

---

## [Decision Letter · Decision Letter 1]

7 Oct 2024

PONE-D-24-25503R1Paw pressure and gait in middle-aged client-owned cats with and without naturally-occurring musculoskeletal diseasePLOS ONE

Dear Dr. Dowgray,

Thank you for submitting your manuscript to PLOS ONE. After careful consideration, we feel that it has merit but does not fully meet PLOS ONE’s publication criteria as it currently stands. Therefore, we invite you to submit a revised version of the manuscript that addresses the points raised during the review process. The manuscript has been taken over from the previous Editor in the R1 phase, by when the necessary revision suggested by reviewer 1 has already been completed.  A second reviewer has been invited to review the manuscript version R1 and according to the suggestions I ask you to carry out an in-depth revision.  I realize that this is an unusual review process, but it appears to improve the manuscript considerably. Please, take into consideration the reviewer's suggestions,especially the one which suggests re-grouping the animals and thus possibly yielding significant results. Please submit your revised manuscript by Nov 21 2024 11:59PM. If you will need more time than this to complete your revisions, please reply to this message or contact the journal office at plosone@plos.org. Please include the following items when submitting your revised manuscript:A rebuttal letter that responds to each point raised by the academic editor and reviewer(s). You should upload this letter as a separate file labeled 'Response to Reviewers'.A marked-up copy of your manuscript that highlights changes made to the original version. You should upload this as a separate file labeled 'Revised Manuscript with Track Changes'.An unmarked version of your revised paper without tracked changes. You should upload this as a separate file labeled 'Manuscript'.

We look forward to receiving your revised manuscript.

Kind regards,

Antal Nógrádi, M.D., Ph.D., D.Sc.

Academic Editor

PLOS ONE

Reviewers' comments:

Reviewer's Responses to Questions

**Comments to the Author**

1. If the authors have adequately addressed your comments raised in a previous round of review and you feel that this manuscript is now acceptable for publication, you may indicate that here to bypass the “Comments to the Author” section, enter your conflict of interest statement in the “Confidential to Editor” section, and submit your "Accept" recommendation.

Reviewer #1: (No Response)

Reviewer #2: (No Response)

2. Is the manuscript technically sound, and do the data support the conclusions?

Reviewer #1: Yes

Reviewer #2: Yes

3. Has the statistical analysis been performed appropriately and rigorously? 

Reviewer #1: Yes

Reviewer #2: Yes

4. Have the authors made all data underlying the findings in their manuscript fully available?

Reviewer #1: Yes

Reviewer #2: Yes

5. Is the manuscript presented in an intelligible fashion and written in standard English?

Reviewer #1: Yes

Reviewer #2: Yes

6. Review Comments to the Author

Reviewer #1: (No Response)

Reviewer #2: This is an interesting paper investigating the possible gait changes in cats with MSD. The experiment is based on the paw pressure and pattern of the animals. The collected data and the statistical tests are appropriate. The authors could not detect significant changes in the aspects of the gait, even with the significant weight differences among the affected and intact animals. It is likely that other aspects of the gait like joint angles and paw lifting abilities would provide more details about the changes of locomotion. The central thoughts of the manuscript are logically established; the authors use English in a most proper way.

There are a few suggestions as follows:

-The authors should consider to redistribute the animals into experimental groups more carefully. Based upon the number and position of the affected limb(s) well-defined groups should be created and each of these groups should be compared to the intact data. This is important because with locomotor diseases compensational processes can strongly distort the collected data depending on the number and sides of the affected limb or limbs. This way the mentioned “intra-cat and intercat” gait variabilities may be avoided, and there would be a chance to detect significant differences in some of the parameters even if “n” of the groups obviously drops this way. Please make clear what groups and subgroups you intend to set up and what plans you intend to make in order to find possible significance between these groups.

-In the Discussion the authors should emphasize more the possible use of other gait measuring methods.

-line 419: Control of the velocity is not necessarily needed if the collected intact data is sorted and characterized by that aspect. By comparing the matching MSD data, different velocities make the assessment even more precise.

7. PLOS authors have the option to publish the peer review history of their article (what does this mean?). If published, this will include your full peer review and any attached files.

Reviewer #1: No

Reviewer #2: No

---

## [Author Response · Author response to Decision Letter 1]

11 Nov 2024

Dear Editor and Reviewer 2, 

Thank you for your time and thoughts in reviewing our paper ‘Paw pressure and gait in middle-aged client-owned cats with and without naturally-occurring musculoskeletal disease’. Your comments and advice are very much appreciated. We have carefully considered all points made and made amendments to the manuscript as a result. These changes, and our responses to your comments, are covered in the point-by-point response that follows:

1. The authors should consider to redistribute the animals into experimental groups more carefully. Based upon the number and position of the affected limb(s) well-defined groups should be created and each of these groups should be compared to the intact data. This is important because with locomotor diseases compensational processes can strongly distort the collected data depending on the number and sides of the affected limb or limbs. This way the mentioned “intra-cat and intercat” gait variabilities may be avoided, and there would be a chance to detect significant differences in some of the parameters even if “n” of the groups obviously drops this way. Please make clear what groups and subgroups you intend to set up and what plans you intend to make in order to find possible significance between these groups.

We have taken into consideration comments about regrouping the cats and further analysis. The sub-setting the cats with musculoskeletal disease into different groups was something we had looked at previously when analyzing the data, but had deliberately excluded as the group sizes tend to be very small. Given the variation in our data (discussed at length in our manuscript), we felt it somewhat redundant to include such small group comparisons. However, as requested, we have now included aspects of this in our resubmission please see, lines 119-120 and 361-373 of our resubmission. The only significance found, at this level between the sub-groups, was that diseased forelimbs had significantly greater forelimb maximum PVF than non-diseased forelimbs; however, this significance disappeared when comparing percentage of total force (to account to some extent for size differences) or when bodyweight was added to the model creating a better fit by removing the MSD groupings. 

2. In the Discussion the authors should emphasize more the possible use of other gait measuring methods.

Thank you for this comment. We have added additional detail in the discussion to include more detail on possible gait measuring methods. Please see lines 431-445 and lines 476 – 486

3. line 419: Control of the velocity is not necessarily needed if the collected intact data is sorted and characterized by that aspect. By comparing the matching MSD data, different velocities make the assessment even more precise.

Further discussion, on the utility of velocity in assessing musculoskeletal disease in cats, has been added in lines 454-455 and 482-485. 

Thank you again for taking the time to review our paper. We hope this resubmission meets the required suggestions from this review process. 

Yours sincerely, 

Dr Nathalie Dowgray and co-authors

---

## [Decision Letter · Decision Letter 2]

14 Nov 2024

Paw pressure and gait in middle-aged client-owned cats with and without naturally-occurring musculoskeletal disease

PONE-D-24-25503R2

Dear Dr. Dowgray,

We’re pleased to inform you that your manuscript has been judged scientifically suitable for publication and will be formally accepted for publication once it meets all outstanding technical requirements.

Kind regards,

Antal Nógrádi, M.D., Ph.D., D.Sc.

Academic Editor

PLOS ONE

Additional Editor Comments (optional):

Reviewers' comments:

Reviewer's Responses to Questions

**Comments to the Author**

1. If the authors have adequately addressed your comments raised in a previous round of review and you feel that this manuscript is now acceptable for publication, you may indicate that here to bypass the “Comments to the Author” section, enter your conflict of interest statement in the “Confidential to Editor” section, and submit your "Accept" recommendation.

Reviewer #2: All comments have been addressed

2. Is the manuscript technically sound, and do the data support the conclusions?

Reviewer #2: Yes

3. Has the statistical analysis been performed appropriately and rigorously? 

Reviewer #2: Yes

4. Have the authors made all data underlying the findings in their manuscript fully available?

Reviewer #2: Yes

5. Is the manuscript presented in an intelligible fashion and written in standard English?

Reviewer #2: Yes

6. Review Comments to the Author

Reviewer #2: The authors have adressed all the raised questions and comments.

The manuscript in its present form is eligible for publication.

7. PLOS authors have the option to publish the peer review history of their article (what does this mean?). If published, this will include your full peer review and any attached files.

Reviewer #2: No

---

## [Editor Report · Acceptance letter]

21 Nov 2024

PONE-D-24-25503R2 

PLOS ONE

Dear Dr. Dowgray, 

I'm pleased to inform you that your manuscript has been deemed suitable for publication in PLOS ONE. Congratulations! Your manuscript is now being handed over to our production team.

Kind regards, 

on behalf of

Prof. Antal Nógrádi 

Academic Editor

PLOS ONE